# Glioblastoma Surgery Imaging—Reporting and Data System: Standardized Reporting of Tumor Volume, Location, and Resectability Based on Automated Segmentations

**DOI:** 10.3390/cancers13122854

**Published:** 2021-06-08

**Authors:** Ivar Kommers, David Bouget, André Pedersen, Roelant S. Eijgelaar, Hilko Ardon, Frederik Barkhof, Lorenzo Bello, Mitchel S. Berger, Marco Conti Nibali, Julia Furtner, Even H. Fyllingen, Shawn Hervey-Jumper, Albert J. S. Idema, Barbara Kiesel, Alfred Kloet, Emmanuel Mandonnet, Domenique M. J. Müller, Pierre A. Robe, Marco Rossi, Lisa M. Sagberg, Tommaso Sciortino, Wimar A. van den Brink, Michiel Wagemakers, Georg Widhalm, Marnix G. Witte, Aeilko H. Zwinderman, Ingerid Reinertsen, Ole Solheim, Philip C. De Witt Hamer

**Affiliations:** 1Department of Neurosurgery, Amsterdam University Medical Centers, Vrije Universiteit, 1081 HV Amsterdam, The Netherlands; i.kommers@amsterdamumc.nl (I.K.); r.eijgelaar@amsterdamumc.nl (R.S.E.); dmj.muller@amsterdamumc.nl (D.M.J.M.); 2Cancer Center Amsterdam, Brain Tumor Center, Amsterdam University Medical Centers, 1081 HV Amsterdam, The Netherlands; 3Department of Health Research, SINTEF Digital, NO-7465 Trondheim, Norway; david.bouget@sintef.no (D.B.); andre.pedersen@sintef.no (A.P.); ingerid.reinertsen@sintef.no (I.R.); 4Department of Neurosurgery, Twee Steden Hospital, 5042 AD Tilburg, The Netherlands; h.ardon@etz.nl; 5Department of Radiology and Nuclear Medicine, Amsterdam University Medical Centers, Vrije Universiteit, 1081 HV Amsterdam, The Netherlands; f.barkhof@amsterdamumc.nl; 6Institutes of Neurology and Healthcare Engineering, University College London, London WC1E 6BT, UK; 7Neurosurgical Oncology Unit, Department of Oncology and Hemato-Oncology, Humanitas Research Hospital, Università Degli Studi di Milano, 20122 Milano, Italy; lorenzo.bello@unimi.it (L.B.); marco.conti@unimi.it (M.C.N.); marco.rossi2@unimi.it (M.R.); tommaso.sciortino@unimi.it (T.S.); 8Department of Neurological Surgery, University of California San Francisco, San Francisco, CA 94143, USA; mitchel.berger@ucsf.edu (M.S.B.); shawn.hervey-jumper@ucsf.edu (S.H.-J.); 9Department of Biomedical Imaging and Image-Guided Therapy, Medical University Vienna, 1090 Wien, Austria; Julia.Furtner@meduniwien.ac.at; 10Department of Circulation and Medical Imaging, Norwegian University of Science and Technology, NO-7491 Trondheim, Norway; even.h.fyllingen@ntnu.no; 11Department of Radiology and Nuclear Medicine, St. Olav’s Hospital, Trondheim University Hospital, NO-7030 Trondheim, Norway; 12Department of Neurosurgery, Northwest Clinics, 1815 JD Alkmaar, The Netherlands; A.J.S.Idema@nwz.nl; 13Department of Neurosurgery, Medical University Vienna, 1090 Wien, Austria; barbara.kiesel@meduniwien.ac.at (B.K.); georg.widhalm@meduniwien.ac.at (G.W.); 14Department of Neurosurgery, Haaglanden Medical Center, 2512 VA The Hague, The Netherlands; a.kloet@mchaaglanden.nl; 15Department of Neurological Surgery, Hôpital Lariboisière, 75010 Paris, France; emmanuel.mandonnet@aphp.fr; 16Department of Neurology and Neurosurgery, University Medical Center Utrecht, 3584 CX Utrecht, The Netherlands; P.Robe@umcutrecht.nl; 17Department of Neurosurgery, St. Olav’s Hospital, Trondheim University Hospital, NO-7030 Trondheim, Norway; lisa.millgard.sagberg@ntnu.no; 18Department of Neurosurgery, Isala, 8025 AB Zwolle, The Netherlands; brink@neurochirurgie-zwolle.nl; 19Department of Neurosurgery, University Medical Center Groningen, University of Groningen, 9713 GZ Groningen, The Netherlands; m.wagemakers@umcg.nl; 20Department of Radiation Oncology, The Netherlands Cancer Institute, 1066 CX Amsterdam, The Netherlands; m.witte@nki.nl; 21Department of Clinical Epidemiology and Biostatistics, Amsterdam University Medical Centers, University of Amsterdam, 1105 AZ Amsterdam, The Netherlands; a.h.zwinderman@amsterdamumc.nl; 22Department of Neuromedicine and Movement Science, Norwegian University of Science and Technology, NO-7491 Trondheim, Norway

**Keywords:** glioblastoma, magnetic resonance imaging, neuroimaging, computer-assisted image processing, machine learning, neurosurgical procedures

## Abstract

**Simple Summary:**

Neurosurgical decisions for patients with glioblastoma depend on tumor characteristics in the preoperative MR scan. Currently, this is based on subjective estimates or manual tumor delineation in the absence of a standard for reporting. We compared tumor features of 1596 patients from 13 institutions extracted from manual segmentations by a human rater and from automated segmentations generated by a machine learning model. The automated segmentations were in excellent agreement with manual segmentations and are practically equivalent regarding tumor features that are potentially relevant for neurosurgical purposes. Standard reports can be generated by open access software, enabling comparison between surgical cohorts, multicenter trials, and patient registries.

**Abstract:**

Treatment decisions for patients with presumed glioblastoma are based on tumor characteristics available from a preoperative MR scan. Tumor characteristics, including volume, location, and resectability, are often estimated or manually delineated. This process is time consuming and subjective. Hence, comparison across cohorts, trials, or registries are subject to assessment bias. In this study, we propose a standardized Glioblastoma Surgery Imaging Reporting and Data System (GSI-RADS) based on an automated method of tumor segmentation that provides standard reports on tumor features that are potentially relevant for glioblastoma surgery. As clinical validation, we determine the agreement in extracted tumor features between the automated method and the current standard of manual segmentations from routine clinical MR scans before treatment. In an observational consecutive cohort of 1596 adult patients with a first time surgery of a glioblastoma from 13 institutions, we segmented gadolinium-enhanced tumor parts both by a human rater and by an automated algorithm. Tumor features were extracted from segmentations of both methods and compared to assess differences, concordance, and equivalence. The laterality, contralateral infiltration, and the laterality indices were in excellent agreement. The native and normalized tumor volumes had excellent agreement, consistency, and equivalence. Multifocality, but not the number of foci, had good agreement and equivalence. The location profiles of cortical and subcortical structures were in excellent agreement. The expected residual tumor volumes and resectability indices had excellent agreement, consistency, and equivalence. Tumor probability maps were in good agreement. In conclusion, automated segmentations are in excellent agreement with manual segmentations and practically equivalent regarding tumor features that are potentially relevant for neurosurgical purposes. Standard GSI-RADS reports can be generated by open access software.

## 1. Introduction

The preoperative MR scan of a patient with a glioblastoma contains essential information that is interpreted by a neurosurgical team for a surgical strategy. Decisions on whether to perform a biopsy or a resection, estimations on how much tumor can be safely removed, the risks of complications and loss of brain functions, and judgements concerning the complexity of the surgery and ensuing pre- and intraoperative diagnostics are imperative for patient outcomes. In addition, the initial scan holds prognostic information, including tumor volume and location [1,2,3], which guides clinical decisions on radiotherapy and chemotherapy and serves patient counseling. In reports of surgical cohorts, multicenter trials, and registries, outcomes are customarily related to measurements of tumor characteristics on the initial scan and related to the outcomes and measurements of other teams [4,5,6,7,8,9,10,11,12,13,14,15]. Furthermore, these reports are pooled in meta-analyses enabling the identification of new patterns in the reported data to guide future clinical decisions [16,17]. Reliable measurements of tumor characteristics are therefore instrumental in patient care and in the development of glioblastoma treatment.

Whereas the response assessment of neuro-oncological treatment mainly focuses on changes in tumor volume over time [18,19] and radiotherapy planning on the clinical target volume on postoperative scans [20,21,22,23], pre-treatment tumor characteristics are of special interest for neurosurgical purposes. In addition to tumor volume, these include measurements of distance to and overlap with brain structures and expected resectability. The current standard is segmentation of the tumor in 3D, while qualitative description, measurement of tumor diameter, and bidimensional products are also in use [24]. These segmentations by human raters have disadvantages. Manual segmentations are time-consuming [25] and therefore expensive. It is common to have inexperienced students or junior investigators as raters for large numbers of segmentations. The level of experience of the rater is an important contributing factor to the accuracy of segmentations [26,27]. Certification of expert raters has not been established. The reproducibility of manual segmentations can be limited, probably due to human error, as attention may fluctuate in monotonous tasks [26,28,29,30,31,32,33]. In addition, segmentation updates or revisions take considerable time.

Automated segmentation algorithms have been developed and compared with manual segmentations as ground truth [34]. Convolutional neural networks [35], in particular employing U-Net [36], dominate the applications. Their performances have been benchmarked on a standardized image dataset (the Brain Tumor Image Segmentation, BraTS [32,34]), using a diagnostic accuracy approach with human rater segmentations as reference. In this approach, the spatial overlap of segmented voxels is typically reported as a Dice score, and the distance of segmentation surfaces as a Hausdorff metric. Nevertheless, this strictly determines the voxel-wise resemblance between an automated segmentation and the reference segmentation. This does not address the clinical utility of these segmentations, and the curated standardized image dataset is not representative for routine scans, which are often of suboptimal quality due to motion artefacts, missing sequences, and other image degradation. Furthermore, in routine scans, brains are not extracted, as is the case in the BraTS dataset.

Standard reporting and data systems (RADS) have been established for several solid tumors, including prostate cancer [37,38], hepatocellular carcinoma [39], head and neck squamous cell carcinoma [40], solitary bone tumors [41], bladder cancer [42], breast cancer [43], lymph node involvement by cancer [44], and lung cancer [45]. These RADS have enabled rules for imaging techniques, terminology for reports, definitions of tumor features, and treatment response, with less practice variation and reproducible tumor classification. Its broad implementation should facilitate collaborations and stimulate evaluation for development and improvement of RADS.

In this study, we determine the agreement in extracted tumor features between automated and manual segmentations from routine clinical MR scans before treatment and describe their discrepancies. We propose a standardized Glioblastoma Surgery Imaging Reporting and Data System (GSI-RADS) to automatically extract tumor features that are potentially relevant for glioblastoma surgery and demonstrate the use of a software module to create standard reports.

## 2. Materials and Methods

### 2.1. Patients and MR Images

We identified all patients of at least 18 years old with a newly diagnosed glioblastoma at first-time surgery between 1 January 2012 and 31 December 2013 from 13 hospitals: Northwest Clinics, Alkmaar, The Netherlands (ALK); Amsterdam University Medical Centers, location VU Medical Center, The Netherlands (AMS); University Medical Center Groningen, The Netherlands (GRO); Medical Center Haaglanden, The Hague, The Netherlands (HAG); Humanitas Research Hospital, Milano, Italy (MIL); Hôpital Lariboisière, Paris, France (PAR); University of California San Francisco Medical Center, US (SFR); Medical Center Slotervaart, Amsterdam, The Netherlands (SLO); St Elisabeth Hospital, Tilburg, The Netherlands (TIL); University Medical Center Utrecht, The Netherlands (UTR); Medical University Vienna, Austria (VIE); and Isala hospital, Zwolle, The Netherlands (ZWO), and between 2007 and 2018 from one hospital: St Olav’s hospital, Trondheim university Hospital, Norway (STO). Patients gave their informed consent for scientific use of their data, as required for each participating hospital. The study was conducted in accordance with the Declaration of Helsinki, and the protocol was approved by the Medical Ethics Review Committee. Data and images for analysis were pseudonymized for analysis.

Patients were identified at each hospital by prospective electronic databases. Part of this cohort was reported earlier to address resectability and comparison of surgical decisions between institutes [46,47]. Descriptive information was collected from the electronical medical records, including age and gender.

Preoperative MR scans were acquired from the hospitals’ archival systems and included a 3D heavily T1-weighted gradient-echo pulse sequence at 1 mm isotropic resolution, obtained before and after administration of intravenous gadolinium, and a T2/FLAIR-weighted gradient-echo pulse sequence. MR scan protocols were standardized in hospitals but not identical between hospitals. Scanners from several vendors were in use, including Siemens, model Sonata, Avanto, Skyra, Prisma and mMR; GE medical systems, model Signa HDxt or DISCOVERY MR750; Toshiba, model Titan3T; and Philips, model Panorama HFO or Ingenuity with field strength of 1.5T or 3T. Detailed scan protocols have been described elsewhere [25,48].

### 2.2. Manual Tumor Segmentations

Tumors were manually segmented in 3D by trained raters using an initiation by either a region growing algorithm [26] (Brainlab SmartBrush, BrainLAB AG, Münich, Germany) or a grow cut algorithm [49] (3D Slicer, http://www.slicer.org, accessed on 3 June 2021) and subsequent manual editing. Trained raters were supervised by neuroradiologists and neurosurgeons. The tumor was defined as gadolinium-enhancing tissue on T1-weighted scans, including nonenhancing enclosed necrosis or cysts.

### 2.3. Automated Tumor Segmentations

A segmentation model was trained following a leave-one-hospital-out cross-validation strategy over the 1596 MRI volumes featured in our dataset, using the AGUNet architecture [50]. The model was trained from scratch, using the Dice Loss as cost function [51] and an Adam optimizer with an initial learning rate of 1e^−3^ and stopped after 30 epochs without validation loss improvement. Data augmentation was performed during training to improve generalization, such as random horizontal and vertical flip, rotation, and translation transforms.

### 2.4. Extracted Tumor Features

To correlate the tumor segmentations with standard anatomy, patient images were nonlinearly registered to a standard anatomical reference space, here consisting of the symmetric Montreal Neurological Institute ICBM2009a atlas, symmetric version 09a (MNI) [52,53], using symmetric image normalization as previously described [54,55]. From both the manual and the automated segmentation of each patient, the following measurements were extracted.

The laterality was defined as the main part of the tumor coinciding with either the left or right hemisphere, or none in the case where a tumor volume was not detected. Contralateral infiltration was defined as binary variable, true if any tumor voxel involved the contralateral hemisphere. The laterality index was defined as an index of tumor distribution between hemispheres, where −1 represents a tumor entirely located in the right hemisphere, 0 represents equal distribution of tumor between both hemispheres, and 1 represents a tumor completely located in the left hemisphere.

The native tumor volume in mL was defined as the number of tumor voxels in patient space times the volume of a tumor voxel in patient space. The normalized tumor volume in mL was defined as the number of tumor voxels in reference space times the volume of a tumor voxel in reference space.

Multifocality was defined as binary variable, true if more than one contrast-enhancing tumor component was observed and the second contrast-enhancing tumor component had a minimum volume of 0.1 mL and a minimum distance between the first and second largest tumor components of 5 mm. The number of foci was counted as the number of unconnected components.

The location profile of cortical structures is represented by the percentage of patients with a tumor per cortical parcel in a circular barplot [56]. We demonstrate the location profile of the cohort for two commonly used brain parcellations, Desikan’s brain parcellation with 96 parcels based on anatomy [57] and Schaefer’s brain parcellation with 17 network classes from 400 parcels based on functional connectivity using a resting state functional MRI [58,59]. Involvement of a patient’s tumor with a parcel was defined as any tumor voxel from a patient overlapping with that parcel.

The location profile of subcortical structures is represented by the percentage of patients with a tumor per white matter structure in a circular barplot [56]. The subcortical white matter structures deemed potentially relevant for surgery comprise a selection of tracts in each hemisphere, consisting of the corticospinal tract with a paracentral and three hand segments; the superior longitudinal fasciculus with three divisions; the arcuate fasciculus with a long, anterior, and posterior segment; the frontal aslant tract; the frontal striatal tract; the inferior fronto-occipital fasciculus; the uncinate fasciculus; the inferior longitudinal fascicle; and the optic radiation. The white matter structure definitions from the Brain Connectivity and Behaviour group were used [60]. The involvement of a patient’s tumor with a structure was defined as any tumor voxel overlapping with that white matter structure.

The expected residual tumor volume and the expected resectability index were calculated with a resection probability map of 451 patients with glioblastoma surgery in the left hemisphere and 464 patients in the right hemisphere, as reference, consisting of a subset of the current study population [46]. To calculate the resectability, the tumor segmentation masked the resection probability map. The resection probabilities of the masked voxels were summed to obtain the expected resectable volume. The preoperative tumor volume minus the expected resectable volume resulted in the expected residual tumor volume in mL. A division of the expected resectable volume by the preoperative tumor volume resulted in the expected resectability index, ranging from 0.0 to 1.0. This method has been detailed and validated elsewhere [46].

The tumor probability map was constructed for the whole population as 3D volume in standard brain space at 1 mm resolution. The fraction of tumors divided by the total number of patients was calculated voxel-wise.

### 2.5. Software Module and Standard Report

The proposed GSI-RADS software (https://github.com/SINTEFMedtek/GSI-RADS, accessed on 3 June 2021) enables the extraction of the described tumor features from a patient’s preoperative MR scan locally. The software has been developed in Python 3 and is compatible for use on Windows 10 (Microsoft Corp., Redmond, WA, USA), macOS (≥10.13; Apple Inc., Cupertino, CA, USA), and Ubuntu Linux 18.04 (Canonical Group Ltd., London, UK). A minimalistic GUI is provided to the user for specifying the required parameters and running the process. The input for the software consists of a 3D T1-weighted gadolinium-enhanced MRI volume provided as a DICOM sequence or NIfTI format. A manual segmentation of the tumor can be provided by the user (e.g., NIfTI format); if not, an automatic segmentation will be generated using the trained model. The output consists of a generated standard report in text (.txt) and CSV format, alongside multiple NIfTI files containing the tumor segmentation as a binary mask (in patient and MNI spaces), the registered MR scan in MNI space, and the anatomical region masks in patient space.

The standard report summarizes the extracted tumor features for each patient. These include the tumor laterality, contralateral infiltration, the laterality index, the native and normalized tumor volumes, the presence of multifocality and the number of foci, the percentage of tumor overlap with cortical parcels and subcortical structures, the expected residual tumor volume and expected resectability, and binary maps of the tumor segmentation in patient space and standard brain space.

### 2.6. Statistical Analysis

Differences in laterality, contralateral infiltration, multifocality, number of foci, and cortical and subcortical profiles between automated and manual segmentations were evaluated in contingency tables and tested for significance of paired data using McNemar’s test for two classes and Friedman’s test for more than two classes. The concordance as a percentage was calculated by dividing the sum of concordant classes over the total number of patients. Differences in native and normalized tumor volumes and expected residual volumes and resectability indices were tested for significance using the Wilcoxon signed-rank test for paired data. Agreement in laterality index, native and normalized tumor volumes, expected residual tumor volumes, and resectability indices between automated and manual segmentations was displayed in histograms, scatter plots, and Bland–Altman plots and calculated as an intraclass-correlation coefficient using a one-way model based on agreement with 95% confidence interval [61,62,63,64]. Equivalence in laterality, contralateral infiltration, native and normalized tumor volumes, multifocality, number of foci, expected residual tumor volumes, and resectability indices were tested using two one-sided tests for the smallest effect size of interest [65]. The smallest effect size of interest for equivalence bounds in proportions was considered to be 10%, for volumes two mL, for foci one focus, and for expected resectability indices 0.1. The product moment correlation coefficient with 95% confidence interval was calculated for the laterality indices, the native and normalized tumor volumes, expected residual tumor volumes, and expected resectability indices between automated and manual segmentations. Voxel-wise agreement was evaluated in tumor probability maps based on automated and manual segmentations. False discovery rates were calculated for the voxel-wise differences using a permutation test, as previously detailed [47,66].

## 3. Results

### 3.1. Patients

A total of 1596 patients were included in this analysis. No scans were excluded based on poor image quality or failed registration. A listing of the populations per hospital is provided in Table 1.

### 3.2. Agreement in Tumor Features between Manual and Automated Segmentations

#### 3.2.1. Laterality, Contralateral Infiltration, and the Laterality Index

The automated and the manual segmentations, respectively, identified 785 (49.2%) and 794 (49.7%) patients with left-sided tumors, 792 (49.6%) and 799 (50.1%) patients with right-sided tumors, and 19 (1.2%) and 3 (0.2%) patients in whom no tumor volume was identified and hence were devoid of laterality, as listed in Table 2. Of the five discordant cases with opposing laterality, four were midline tumor with slightly dissimilar tumor voxel numbers in either hemisphere, and one scan was of poor quality with faint gadolinium enhancement of the tumor that the automated method failed to detect while a false positive segmentation of choroidal plexus was segmented contralaterally. In 17 (1.1%) patients, the automated segmentation did not identify a tumor, whereas the human rater did, due to minute tumor size, faint gadolinium enhancement, or poor scan quality. The observed laterality difference was statistically not different from zero (odds ratio: 0.98, 95% CI: 0.89–1.09; *p*-value = 0.744) and statistically equivalent to zero (95% CI: −0.029 to 0.030; Z = −5.59, *p*-value < 0.0001). The concordance was 98.6%.

Contralateral infiltration was observed in 430 (26.9%) patients based on the automated segmentations and in 469 (29.4%) based on the manual segmentations, as listed in Table 3. The observed difference in contralateral infiltration was statistically not different from zero (Z = 1.54, *p*-value = 0.125) and statistically not equivalent to zero (95% CI: −0.007 to 0.056; Z = −1.61, *p*-value = 0.0541). The concordance was 95.4%.

The distribution of the laterality indices determined by automated and manual segmentations and their correlation are shown in Figure 1A and the Bland–Altman plot in Figure 1B. The correlation coefficient was 0.998 (95% CI: 0.998–0.998). No bias was observed (0.00039, 95% CI: −0.0024 to 0.0032). The lower and upper 95% limits of agreement were −0.11 and 0.11.

This indicates excellent agreement to detect laterality, contralateral infiltration, and the laterality index between the segmentation methods.

#### 3.2.2. Tumor Volumes

The difference between the native and normalized tumor volumes was plotted in Figure 2A,B. The median (interquartile range) of this difference for automated segmentations was −2.6 (6.8) mL and for manual segmentations −3.2 (7.5) mL. Apparently, the standard brain is somewhat larger than the brains of many patients. Therefore, we assessed normalized tumor volume in addition to native tumor volume.

The median (interquartile range) of the native tumor volumes was 26.5 (36.6) mL for automated segmentations and 26.6 (37.1) mL for manual segmentations, with a small but clinically negligible difference (0.4 mL, 95% CI: 0.4–0.5; *p*-value < 0.0001), well within the smallest effect size of interest of 2 mL (one-sided test for the upper bound t = −11.4, df = 1595, *p*-value < 0.0001 and for the lower bound t = 17.3, df = 1595, *p*-value < 0.0001).

The median (interquartile range) of the normalized tumor volumes was 30.1 (42.4) mL for the automated segmentations, and 31.2 (42.0) mL for the manual segmentations, again with a negligibly small difference (1.0 mL, 95% CI: 0.8–1.1; *p*-value < 0.0001), well within the smallest size of interest of 2 mL (one-sided test for the upper bound t = −4.9, df = 1595, *p*-value < 0.0001 and for the lower bound t = 17.3, df = 1595, *p*-value < 0.0001).

The intraclass correlation coefficient of the native tumor volumes was 98.2% (95% CI: 98.0–98.3%) and of the normalized tumor volumes 97.9% (95% CI: 97.7–98.1%), indicating excellent internal consistency.

In Figure 2C,E, the native and normalized tumor volumes based on automated and manual segmentations are plotted, indicating excellent agreement. In the Bland–Altman plots in Figure 2D,F, a small negligible systematic bias was observed between the automated and manual segmentations for native (0.4 mL, 95% CI: 0.1–0.7) and normalized tumor volumes (1.2 mL, 95% CI: 0.9–1.5). The limits of agreement were between −11.0 and 11.3 mL for the native tumor volumes and between −11.8 and 14.2 mL for the normalized tumor volumes.

This indicates excellent agreement, consistency, and equivalence in native and normalized tumor volume measurements between the automated and manual segmentations.

#### 3.2.3. Multifocality and Number of Foci

Multifocality was identified in 320 (20.1%) patients for automated segmentations and in 374 (23.4%) for manual segmentations, as listed in Table 4. The observed multifocality difference was statistically different from zero (odds ratio 0.51, 95% CI: 0.37 to 0.72; *p*-value < 0.0001) and statistically equivalent to zero (95% CI: 0.010 to 0.058; Z = −4.54, *p*-value < 0.0001). The concordance was 89.5%.

The number of foci as determined by automated and manual segmentations is listed in Table 5. The observed number of foci was statistically different (Friedman chi-squared = 40.3, *p*-value < 0.0001). The concordance was 83.8%.

This indicates good agreement and equivalence in multifocality, but not in the number of foci between the automated and manual segmentations.

#### 3.2.4. Location Profile of Cortical Parcels

The location profiles of the 96 cortical parcels from Desikan’s brain parcellation for the patient population are shown in Figure 3A,B according to the manual and automated segmentations. The well-known preferred locations of glioblastoma are apparent, and the incidence profiles of cortical involvement are almost identical between the segmentation methods. The correlation coefficient of the number of patients with parcel involvement as displayed in Figure 3C was 0.999 (95% CI: 0.999–0.999).

This indicates excellent agreement.

The location profiles of the 400 cortical parcels converging into 17 network classes from Schaefer’s brain parcellation for the patient population are shown in Figure 4A,B for the manual and automated segmentations. The incidence profiles of cortical involvement are almost identical between the segmentation methods. The correlation coefficient of the number of patients with parcel involvement as displayed in Figure 4C was 0.998 (95% CI: 0.998–0.999).

This indicates excellent agreement in cortical incidence profiles between the segmentation methods.

#### 3.2.5. Location Profile of Subcortical Structures

The location profiles of 17 white matter tracts in either hemisphere for tumor overlap were compared for the whole population between the automated and manual segmentations in Figure 5A,B, respectively. The incidence profiles of cortical involvement are almost identical between the segmentation methods. The correlation coefficient of the number of patients with tract involvement was 0.999 (0.999–1.000), as displayed in Figure 5C.

This indicates excellent agreement between the segmentation methods.

#### 3.2.6. Expected Residual Tumor Volume and Expected Resectability Index

The median (interquartile range) of the expected residual tumor volume was 4.5 (7.2) mL for automated segmentations and 4.7 (7.5) mL for manual segmentations, which have a small clinically negligible difference (0.2 mL, 95% CI: 0.2–0.3; *p*-value < 0.0001), within the smallest effect size of interest of 2 mL (one-sided test for the upper bound t = −35.6, df = 1575, *p*-value < 0.0001 and for the lower bound t = 56.7, df = 1575, *p*-value < 0.0001).

The median (interquartile range) of the expected resectability index was 0.857 (0.099) for automated segmentations and 0.849 (0.098) for manual segmentations, which have a small clinically negligible difference (−0.0025, 95% CI: −0.0035 to −0.0020; *p*-value < 0.0001), within the smallest effect size of interest of 0.1 (one-sided test for the upper bound t = −125, df = 1575, *p*-value < 0.0001 and for the lower bound t = 112, df = 1575, *p*-value < 0.0001).

Between automated and manual segmentations, the intraclass coefficient of the expected residual tumor volumes was 96.5% (95% CI: 96.2–96.8%), and the expected resectability index was 94.2% (95% CI: 93.6–94.7%), indicating excellent consistency.

In Figure 6A,C, the expected residual tumor volume and resectability index are plotted, indicating excellent correlation between the automated and manual segmentations. In the Bland–Altman plots in Figure 6B,D, a small negligible bias was observed between the automated and manual segmentations for the expected residual tumor volume (0.5 mL, 95% CI: 0.4–0.5) and for the expected resectability index (−0.005, 95% CI: −0.004 to −0.007). The limits of agreement were between −2.9 and 3.8 mL for the expected residual tumor volume and between −0.07 and 0.06 for the expected resectability index.

This indicates excellent agreement, consistency, and equivalence in expected residual tumor volume and resectability index between the segmentation methods.

#### 3.2.7. Tumor Probability Map

The tumor probability maps based on automated and manual segmentations are provided in Figure 7. The maps were almost identical. Of 1.9 million brain voxels, none had an incidence difference with a false discovery rate below 20%.

This indicates excellent tumor probability map agreement between the segmentation methods.

### 3.3. Examples of Disagreement between Manual and Automated Segmentations

From inspection of the cases that showed lower agreement between automated and manual segmentations, four categories of disagreement emerged, as demonstrated in Figure 8: (i) false negative cystic tumor portions in the automated segmentations; (ii) false negative enhancing tumor volume, typically satellite lesions missed by the automated segmentation; (iii) mismatch in inclusion of nonenhancing tumor portions; and (iv) false positive vasculature structures or choroidal plexus, mistaken for a tumor.

### 3.4. GSI-RADS Software and Standard Report

An example of the generated output is shown in Figure 9 The numerical results are displayed as text in a window and can be exported in csv file format.

## 4. Discussion

The main finding of this study is that automated segmentations are in excellent agreement with manual segmentations regarding extracted tumor features, such as laterality, tumor volume, multifocality, location profiles of cortical parcels and subcortical structures, resectability, and tumor probability maps, which are potentially relevant for neurosurgical planning and reporting. This agreement supports at least equal validity of automated segmentations for these purposes. The generation of automated segmentations is more rapid and more reproducible than manual segmentations, as previously demonstrated [27]. We propose to substitute manual delineations with automated segmentation methods as standard in reports of patients with glioblastoma. To facilitate the distribution of these standard methods, we provide GSI-RADS as software to extract the most relevant tumor features from an MR scan, consisting of tumor laterality, volume, multifocality, location profiles of cortical and subcortical involvement, and resectability.

The use of a uniform method by the neurosurgical community to delineate a tumor and to extract tumor features would be an important step towards standardization across studies and between neurosurgical teams. A suitable segmentation method for neurosurgical use has several requirements: the method should be user friendly, rapid, scalable, accurate, reproducible, affordable, and valid [67]. The present software module is designed to minimize user interaction to import the DICOM scan. The processing duration of the automated method is a fraction of the manual method, which typically takes 30 min per patient [27], deterring to scale to cohorts larger than a few hundred patients. In absence of a ground truth for the exact tumor location, the accuracy of either method remains undetermined. Histopathological and molecular determination of tumor presence based on detailed multiregion sampling would theoretically be the ultimate ground truth [68]. This is infeasible for a patient cohort for obvious reasons. A second-best ground truth is postmortem investigation, although this would restrict a correlation to a recent last scan, and results may not extrapolate to the early stage of disease. An alternative ground truth could be an ensemble of segmentations by multiple expert raters, but this takes considerable time and expense restricted to a limited numbers of patients [69]. Therefore, we took a pragmatic approach and with equivalence between the segmentation methods, the question on the better method can remain unanswered. Automated segmentations are entirely reproducible and free, providing segmentations that can be updated through batch processing, whereas human raters are subject to disagreement between and within raters, yielding unreproducible data from a task that is not trivial in time and expense. In this study, we demonstrate that automated segmentations are equivalent to manual segmentations regarding neurosurgical tumor characteristics, hence they are equally valid. Either segmentation method may yield questionable results in a small subset of atypical tumors, characterized by faint contrast enhancement with large nonenhancing tumor portions, large cysts, or image artefacts. In the absence of a ground truth, we would argue that the reproducibility of an automated segmentation is preferable over arduous manual assessment, even in such less well-defined cases. Likewise, a pragmatic and reproducible standard for tumor volume, focality, location, and resectability based on automated segmentation is preferable over manual delineation.

Our finding that an automated processing by a ‘machine’ can replace a tedious and error-prone task by a ‘human’ adds to an already long list [70,71,72,73,74]. From this perspective, our findings are unsurprising and fit in the development of successful implementations of processes automated by deep learning.

Thus far, no other applications have been developed to extract tumor characteristics for use in glioblastoma surgery, although several applications were developed to segment the tumor in scans. The Brain Tumor Image Analysis tool (BraTumIA) has been developed to segment three brain tumor compartments using four scan sequences [33,75] and has been shown to have good agreement with manual tumor volumes on preoperative scans. The Pearson’s correlation coefficient between manual and BraTumIA tumor volumes was 0.8 based on 19 patients [75] and 0.88 based on 58 patients [76], albeit with a systematic overestimation. In addition, the BraTS challenge has been held yearly since 2012, which aims to improve disease diagnosis, treatment planning, monitoring, and clinical trials by means of reliable tumor segmentation. Participants have applied more than 200 models over the years. Many models were updated versions of previous submissions. As far as we are aware, none of these models has been used to generate tumor characteristics for neurosurgical practice. Therefore, the quest for the best performance in a common dataset by ranking of Dice score is not necessarily representative for clinical practice. In this study, we sought to address whether automation could replace manual labor without compromising validity in terms of tumor features and to make the software readily available for others to use and validate further, both clinically and technically. Future improvements of automated methods can be easily integrated in updated software.

A strength of this study is good external validity given the mixture of institutions, scanners, scan protocols, and patients. Until standardized scan acquisition protocols are implemented in neuro-oncological care [77], automated segmentation methods should resolve this practice variation. Another strength is the relatively large dataset for training the automated method. A limitation is that we used manual segmentations from one trained rater per tumor, although this probably represents current practice in neurosurgical reports of tumor characteristics.

A practical implication is that standard reports for glioblastoma surgery can now be generated by GSI-RADS. Obviously, improved patient outcomes cannot be expected from better reporting in itself. Indirectly, improved outcomes may result from more accurate data-driven decisions on the use of preoperative techniques such as DTI-based tractography, functional MRI, transcranial stimulation, and intraoperative stimulation mapping. Another indirect effect may be the facilitation of consultation between neurosurgeons and teams and possibly in referral patterns by better recognition of complex surgical cases regarding tumor location and eloquence. An example would be the identification of a more complex tumor near the arcuate fascicle, for instance, by a lower expected resectability index and infiltration of this tract, indicating additional preoperative diagnostics to detail the relation between the tumor and the tract and the use of intraoperative stimulation mapping to safely maximize tumor removal. As such, the automated methods hold potential for development of a quantitative standard for eloquence. Reliable definitions of pretreatment tumor characteristics from MR scans may also facilitate less biased comparisons across institutions, studies, or quality registries. Furthermore, prognostic information, surgical treatment evaluation, and response assessment may indirectly improve the risk stratification of patient cohorts. Finally, the standardized reports could speed up the learning curve and serve in the education and training of neurosurgeons.

In future efforts, several directions are important to explore. The automated segmentations can be extended to other pathology, such as lower-grade nonenhancing glioma, brain metastasis, and meningioma. Alternative automated methods can be benchmarked against the current results. The presented automated method can be trained with data from additional patients and institutions. New tumor features will be added to the standard report, such as different aspects of multifocality and the infiltration and disconnection of white matter pathways. These new measures should be compared with patient outcomes for evaluation of their clinical use [46]. This may, for instance, result in a quantitative assessment of risk for surgical complications and risk for early tumor progression. Other tumor compartments can be included, such as the T2/FLAIR hyperintense region, necrotic or ischemic tissue, hemorrhage, cyst fluid, and ultimately molecular heterogeneity and metabolic activity. In addition, reliable tumor segmentations over time and at different stages of disease would be instrumental to provide standardized reports of postsurgical evaluation and treatment response assessment. Finally, distribution of the software should be available for multiple platforms and environments, such as a standalone web-based application.

## 5. Conclusions

Automated segmentations are in excellent agreement with manual segmentations and are practically equivalent regarding tumor features that are potentially relevant for neurosurgical purposes. A standard GSI-RADS report is proposed for these tumor features, including the laterality, volume, multifocality, location, and resectability (https://github.com/SINTEFMedtek/GSI-RADS, accessed on 3 June 2021).

## Figures and Tables

**Figure 1 cancers-13-02854-f001:**
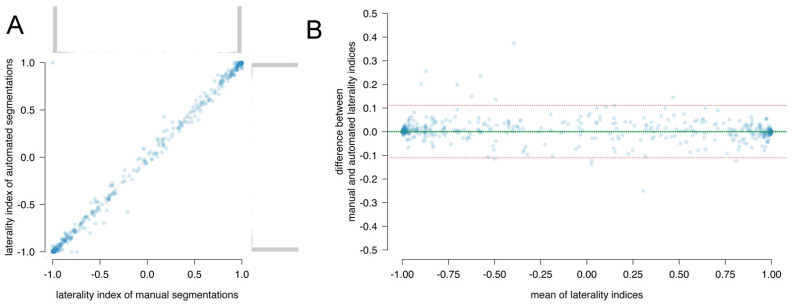
Comparison between automated and manual segmentations for the laterality index in (**A**) a correlation plot with histograms in the margin and (**B**) a Bland–Altman plot. In the scatterplots, each dot represents the laterality indices of one patient. The diagonal indicates the identity line. The Bland–Altman plots of the mean of laterality indices versus the difference between the laterality indices. Each dot represents one patient. The bias is plotted as solid green line with 95% CI as dotted green lines. The limits of agreement are plotted as dotted red lines.

**Figure 2 cancers-13-02854-f002:**
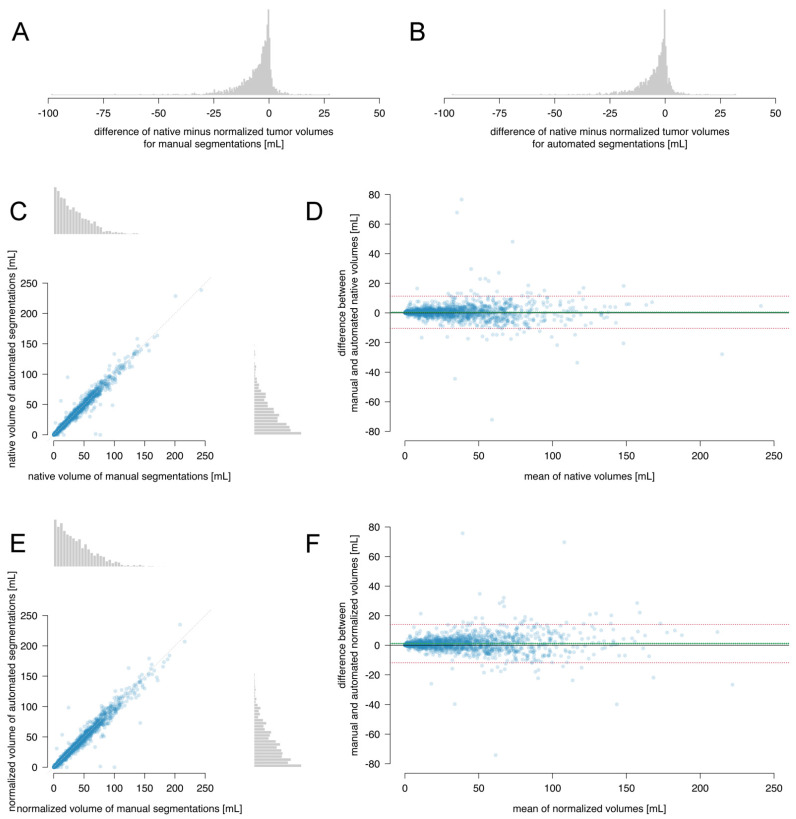
Comparison between manual and automated segmentations: (**A**) a histogram of absolute difference for the manual segmentations and (**B**) for the automated segmentations, (**C**) a correlation plot of the native tumor volumes with histograms in the margin, (**D**) a Bland–Altman plot for the native tumor volumes, (**E**) a correlation plot of the normalized tumor volumes with histograms in the margin, and (**F**) a Bland–Altman plot for the normalized tumor volumes. Each dot represents the volumes of one patient. The dotted diagonal in (**C**,**E**) indicates the identity line. The bias is plotted as solid green line with 95% CI as dotted green lines and the limits of agreement as dotted red lines in (**D**,**F**).

**Figure 3 cancers-13-02854-f003:**
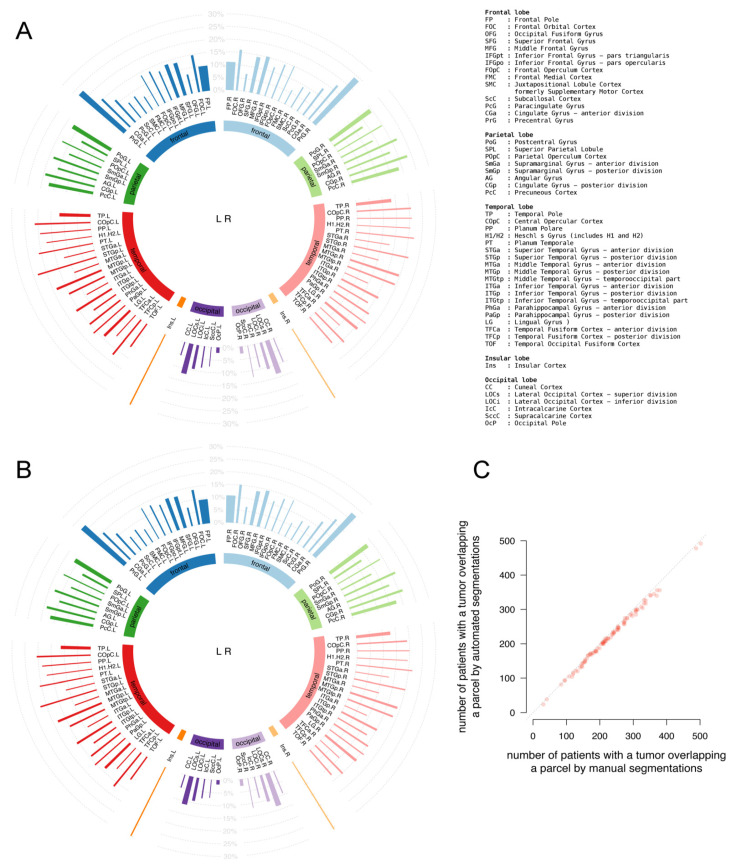
Comparison of tumor location profiles of cortical structures between (**A**) automated and (**B**) manual segmentations in Desikan’s brain parcellation. In the circular bar plots, each bar represents one parcel from the Desikan’s brain parcellation categorized by lobe. Abbreviations are referring to anatomical parcels as detailed in the legend. The height of a bar represents the percentage of patients, indicated in grey, with tumor involvement in a parcel. The width of a bar corresponds with the relative volume of a parcel. (**C**) Correlation plot between the number of patients with parcel involvement between the manual and automated segmentations. The dotted diagonal indicates the identity line.

**Figure 4 cancers-13-02854-f004:**
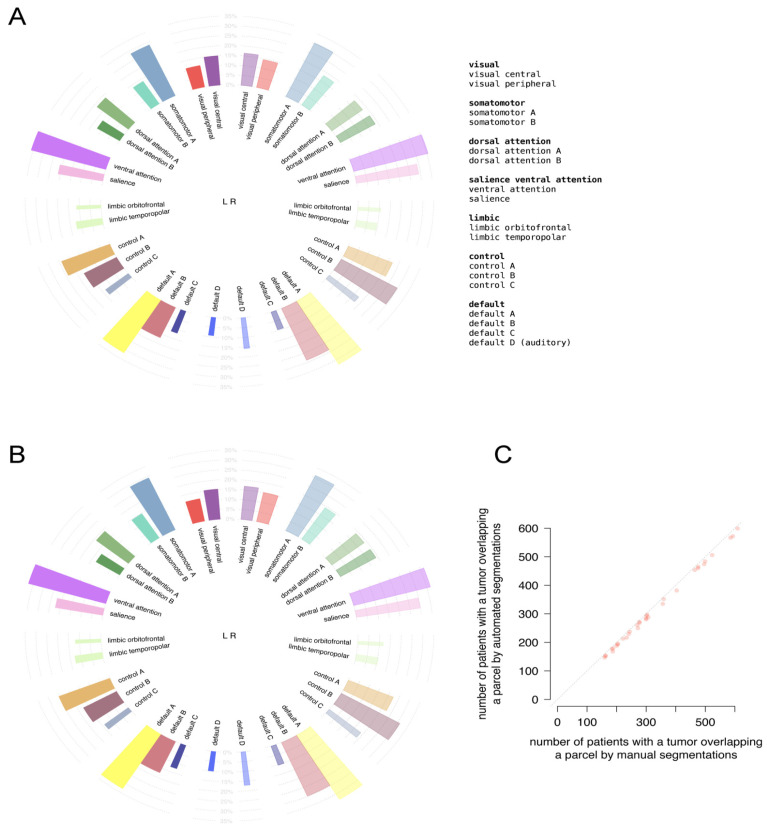
Comparison of tumor location profiles of cortical structures between (**A**) automated and (**B**) manual segmentations in Schaefer’s brain parcellation. In the circular bar plots, each bar represents one network class parcel. Abbreviations refer to classes as detailed in legend. The height of a bar represents the percentage of patients, indicated in grey, with tumor involvement in a class parcel. The width of a bar corresponds with the relative volume of a parcel. (**C**) Correlation plot between the number of patients with parcel involvement between the manual and automated segmentations. The dotted diagonal indicates the identity line.

**Figure 5 cancers-13-02854-f005:**
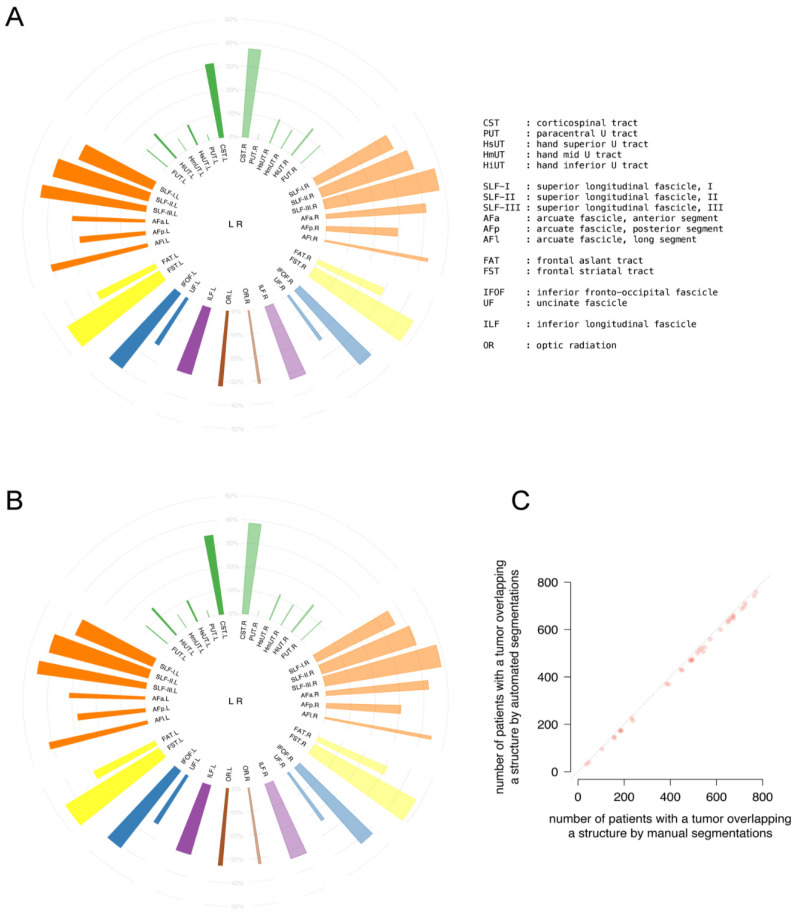
Comparison of tumor location profiles of subcortical white matter structures between (**A**) automated and (**B**) manual segmentations. In the circular bar plots, each bar represents one tract or tract segment. Abbreviations refer to structures as detailed in legend. The height of a bar represents the percentage of patients with tumor involvement in a structure indicated in grey. The width of a bar corresponds with the relative volume of a structure. (**C**) Correlation plot between the number of patients with structure involvement between the manual and automated segmentations. The dotted diagonal indicates the identity line.

**Figure 6 cancers-13-02854-f006:**
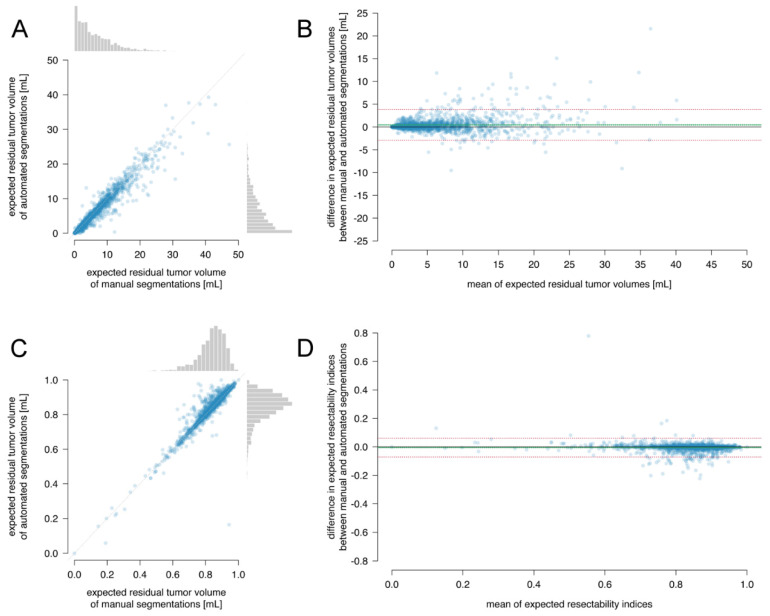
Comparison between automated and manual segmentations for (**A**,**B**) expected residual tumor volumes and (**C**,**D**) expected resectability index. In the scatterplots (**A**,**C**), each dot represents the data of one patient. The diagonal indicates the identity line. The boxplots display the distributions with median, 25% and 75% quartiles as hinges and 1.5 times the interquartile distance as whiskers. The Bland–Altman plots (**B**,**D**) of the mean of expectations versus the difference between expectations. Each dot represents one patient. The bias is plotted as solid line with 95% CI as dotted lines. The limits of agreement are plotted as dashed lines.

**Figure 7 cancers-13-02854-f007:**
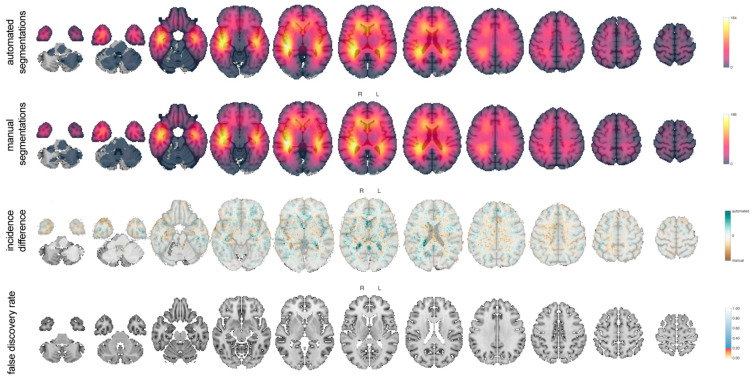
Tumor probability maps for the automated and manual segmentations. Each voxel represents the tumor incidence in the study population with false discovery rates of the difference between the incidences, as specified in the legend.

**Figure 8 cancers-13-02854-f008:**
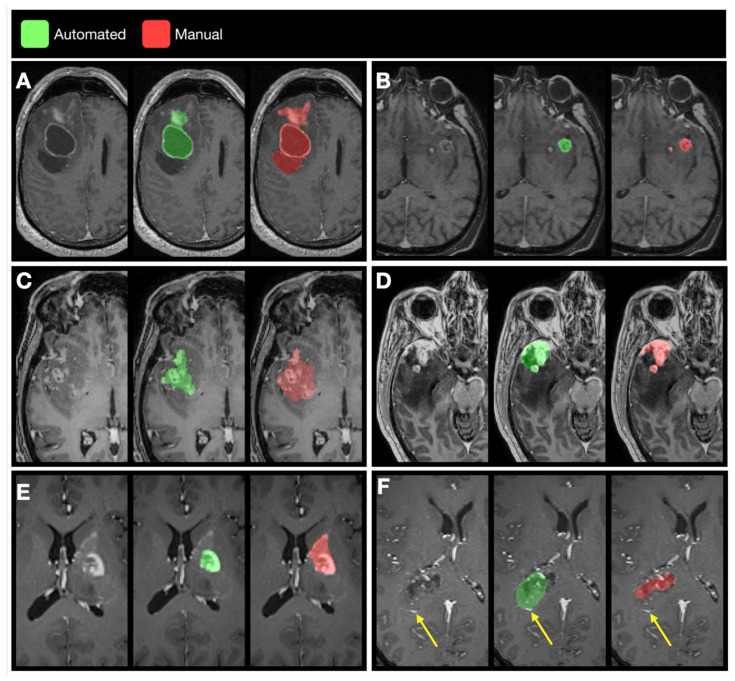
Examples of disagreement between manual and automated segmentation. (**A**) The automated and manual segmentation both included the tumor cyst with enhancing rim, but the automated segmentation did not include the cyst without enhancing rim nor some of the solid tumor extensions, as examples of false negative cyst detection. (**B**) The automated segmentation did not include the satellite lesion, as example of false negative enhancing tumor component. (**C**) The manual segmentation included tissue as tumor portions, whereas the automated segmentation did not, as example of either false positive inclusion of nonenhancing tissue by the manual segmentation or false negative exclusion of nonenhancing tumor exclusion by the automated segmentation. (**D**) Conversely, the manual segmentation excluded tissue from the tumor compartment, whereas the automated segmentation included this tissue, as example of either false negative exclusion of nonenhancing tissue by the manual segmentation or false positive inclusion of nonenhancing tumor exclusion by the automated segmentation. (**E**) The manual segmentation included a vascular structure, as example of a false positive vasculature structure. (**F**) Conversely, the automated segmentation included a vascular structure indicated by the yellow arrow, as another example of a false positive finding.

**Figure 9 cancers-13-02854-f009:**
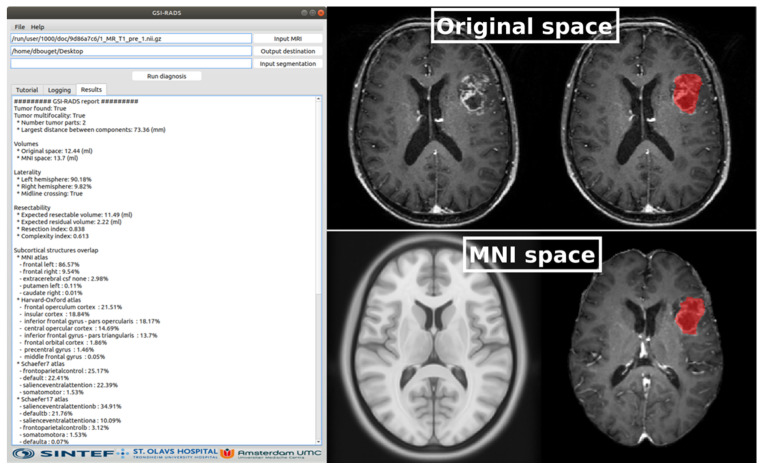
Illustration of the GSI-RADS software and standard report. At the left, the standard report is displayed in text format. At the top right, the patient MRI scan and the patient MRI scan with overlayed automated tumor segmentation are displayed, and at the bottom right, the standard brain space and the registered patient MRI scan with overlayed automated tumor segmentation in standard brain space are demonstrated.

**Table 1 cancers-13-02854-t001:** Patient characteristics.

**Hospital**	**NWZ**	**SLZ**	**ISALA**	**PARIS**	**HUM**	**MUW**	**UMCG**
*n*	38	49	72	74	75	83	86
females, *n* (%)	13 (34.2%)	25 (51.0%)	9 (12.5%)	33 (44.6%)	29 (38.7%)	36 (43.4%)	31 (36.0%)
median age in years (interquartile range)	63.4 (17.4)	63.6 (14.2)	67.2 (20.7)	59.0 (13.5)	62.7 (16.3)	67.3 (19.7)	62.8 (12.4)
**Hospital**	**VUmc**	**HMC**	**UCSF**	**ETZ**	**UMCU**	**STO**	**overall**
*n*	97	103	134	153	171	461	1596
females, *n* (%)	35 (36.1%)	38 (36.9%)	49 (36.6%)	50 (32.7%)	63 (36.8%)	189 (41.0%)	600 (37.6%)
median age in years (interquartile range)	64.0 (16.2)	61.1 (18.1)	64.2 (14.8)	63.8 (12.2)	66.2 (16.4)	61.7 (14.4)	63.2 (15.7)

**Table 2 cancers-13-02854-t002:** Contingency table of laterality between automated and manual segmentations.

	Laterality by Automated Segmentation			
**Laterality by Manual Segmentation**	left	right	none	subtotal
left	782	2	10	794
right	3	789	7	799
none	0	1	2	3
subtotal	785	792	19	

**Table 3 cancers-13-02854-t003:** Contingency table of contralateral infiltration between automated and manual segmentations.

	Contralateral Infiltration by Automated Segmentation		
**Contralateral Infiltration by Manual Segmentation**	no	yes	subtotal
no	1110	17	1127
yes	56	413	469
subtotal	1166	430	1596

**Table 4 cancers-13-02854-t004:** Contingency table of multifocality between automated and manual segmentations.

	Multifocality by Automated Segmentation		
**Multifocality by Manual Segmentation**	no	yes	subtotal
no	1165	57	1222
yes	111	263	374
subtotal	1276	320	1596

**Table 5 cancers-13-02854-t005:** Contingency table of multifocality between automated and manual segmentations.

	Number of Foci by Automated Segmentation						
**Number of Foci by Manual Segmentation**	0	1	2	3	4	5	subtotal
0	2	1	0	0	0	0	3
1	13	1149	52	5	0	0	1219
2	4	86	148	16	1	0	255
3	0	19	32	36	2	0	89
4	0	1	8	7	3	1	20
5	0	1	2	2	2	0	7
6	0	0	0	0	1	0	1
7	0	0	0	0	0	1	1
11	0	0	0	0	0	1	1
subtotal	19	1257	242	66	9	3	1596

## Data Availability

The manual segmentation data can be found as a publicly archived dataset (https://doi.org/10.17026/dans-xam-j5aw, accessed on 3 June 2021). The data and code for analysis can be found as a public archive (https://gitlab.com/picture/gsi-rads, accessed on 3 June 2021). The open access software can be found as a public archive (https://github.com/SINTEFMedtek/GSI-RADS, accessed on 3 June 2021).

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
