# Peer review of "Glioblastoma Surgery Imaging—Reporting and Data System: Standardized Reporting of Tumor Volume, Location, and Resectability Based on Automated Segmentations"

_cancers, 2021, doi:10.3390/cancers13122854_

Round 1

Reviewer 1 Report

In this study, the authors investigated the agreement of extracted tumor features between automated and manual segmentations of gadolinium-enhanced tumor parts from routine clinical MR scans of 1596 glioblastoma patients from 13 institutions.

Furthermore, the authors propose a standardized Glioblastoma Surgery Imaging Reporting and Data System (GSI-RADS) in order to automatically extract tumor features that are potentially relevant for glioblastoma surgery.

This is an interesting and relevant paper about an important topic: introduction of a standardized preoperative glioblastoma assessment system. However, further evaluations with more appropriate methods are required as well as methods and results need revision. Therefore I recommend major revision.

Comments:

Abstract: A clear statement of the purpose of the study is required. The title suggests that the introduction of GSI-RADS is the purpose of the study. However, in the abstract the authors state that the determination of the agreement in extracted tumor features between automated and manual segmentations from routine clinical MR scans is the purpose. Of course, this is important for an automatic classification system of glioblastoma. However, investigation of the agreement between automated and manual segmentations was performed in several previous studies. On the other hand, the introduction of GSI-RADS is innovative and should be emphasized more.

Methods:

  1. Which software packages were used for manual segmentation, i.e. for region growing and grow cut, respectively?
  2. Is the GSI-RADS software available for the community? If so, please provide a link for download as reference.
  3. Which information is included in the standard report? How is this report and the information in it relevant for surgical planning, treatment planning, or patient management. The authors propose standardized GSI-RADS but did not either describe the content and the structure of the report or the potential importance of it.
  4. I wonder why the authors didn't introduce a classification similar to BIRADS, i.e. a score between 1 to 6 or similar. Is this an option for GSI-RADS. Please comment.
  5. Line 266: Calculation of the accuracy requires information of a gold standard. This is not available when comparing automated and manual segmentations. In my opinion, the percentage that was calculated here simply was the percentage of concordance but not the accuracy. Moreover, other statistical methods are available that are better suited for this purpose: intraclass correlation coefficient (ICC) and Lin’s concordance correlation coefficient (CCC) for the features, and the Dice score for the segmentation methods. The authors should include the evaluations in a revised version of their manuscript.

Results: The font size in the Figures (especially Fig. 4) is too small. It was even impossible for me to read the text even with a magnifying glass. The authors should revise the Figures accordingly.

Reviewer 2 Report

 the study represents a good starting point for further studies on the subject

Reviewer 3 Report

This is a well described and documented work concerning a major issue in HGG surgery.

In this paper the Authors show us a novel method able to do the  pre op volumetric assessment of the lesion in an automatic and standardized way in order to avoid all of those "human subjective components" that could potentially lead to mistakes in evaluation.

I think that this is an interesting and well documented paper and it could be suitable for publication in your journal.

Round 2

Reviewer 1 Report

The authors answered my questions satisfactorily. All proposed changes have been made. I have no further concerns and recommend acceptance of the manuscript.